# Cylindrical Cavity Sensor for Distinction of Various Driveability Index Gasoline with Temperature Robustness

**DOI:** 10.3390/s19214626

**Published:** 2019-10-24

**Authors:** Chong Hyun Lee, Yoon-Sang Jeong, Hina Ashraf

**Affiliations:** 1Department of Ocean System Engineering, Jeju National University, Jeju 690-756, Korea; jyssolt@naver.com (Y.-S.J.); hina.ashraf@nu.edu.pk (H.A.); 2Department of Electrical Engineering, National University of Computer and Emerging Sciences, Foundation for Advancement of Science and Technology (NUCES-FAST), H 11/4, Islamabad 44000, Pakistan

**Keywords:** driveability index (DI), normal gasoline, permittivity, sensitive microwave sensor

## Abstract

In this paper, a cylindrical cavity sensor based on microwave resonant theory is proposed to distinguish between various driveability index gasolines under temperature variations. The working principle of the proposed sensor is based on the fact that the change in permittivity of gasoline samples inside cavity sensor will also cause a change in resonant frequency. The proposed sensor has good sensitivity in terms of resonant frequency separation, which enables it to capture the minute permittivity changes and distinguish different gasolines. By using a normal gasoline permittivity of 2.15 and changing sensor dimension parameters, the sensor was designed by high-frequency structure simulator (HFSS). The designed sensor has a resonant frequency of 7.119 GHz for the *TM*_012_ mode with a 19.2 mm radius, a 35 mm height, and one-port coupling probe of 8 mm height. The proposed cylindrical cavity sensor shows advantages of excellent resonant characteristics of small cavity size and small sample amount. To optimize and verify the parameters of the sensor, many experiments have been carried out using HFSS and a vector network analyzer (VNA). Consequently, the proposed sensor is proven to be robust to temperature changes in terms of resonant frequency separation. The minimum frequency separation to distinguish gasoline samples is found to be larger than 29 MHz with reflection coefficients under −11 dB for temperature changes from −35 °C to 0 °C. The consistency of experimental and theoretical results also are presented, which guarantees accuracy of the sensor for the distinction of gasoline.

## 1. Introduction

Gasoline is a complex combination of chemical components used as fuel for spark–ignition engines [1]. The performance of gasoline over the entire range of operating temperatures is measured by the driveability index (DI) [2,3]. A parametric study on fuel has shown that DI is closely related to fuel volatility and has a significant impact on engine cold start performance [4]. DI is a measure of fuel volatility that is a function of ethanol content and temperature ranges at which 10% (T10), 50% (T50), and 90% (T90) of the gasoline boils. The DI of fuel purchased is correlated with ambient temperature. The vehicles that use different blends of gasoline require some form of sensors to measure the DI and composition of the fuel. For such measurements, highly accurate and sensitive sensors are required, however the traditional sensor performance is insufficient to meet these criteria. At present, different techniques have been used to measure the DI of fuel. Lambert et. al. estimates the DI of the sample by heating a fuel sample placed between two narrow parallel plates of a capacitor, and then measuring the change in capacitance of the sensing element as a function of time and temperature [5]. Another reported technique utilized platinum resistance temperature detectors (RTDs) to compare the engine exhaust gas temperatures during cold starts to actual engine exhaust air/fuel ratios, using low and high DI fuels to detect the cold start volatility characteristic of the fuel [6]. The limitation to this technique is that it can not differentiate between various HiDI but only detect the presence of a HiDI fuel. Moreover, both the techniques mentioned above generally require the sample to be heated for a substantial amount of time before the sensor becomes effective for control following a cold start. By contrast, RF techniques do not require the sample to be heated so quick measurements are possible.

The gasoline driveability index (DI) sensor is used when starting a vehicle. It should be small in size for mounting in the vehicle and for fast measurements. Conventional analysis technology based on gas chromatography/mass spectrometry can analyze samples with high resolution, but this process is complicated and time-consuming [7,8]. In comparison, radio frequency and microwave technologies are fast and simple processes. Therefore, they are efficient and cost-effective approaches for DI sensing. Recently, many researches have proposed microwave-based techniques to identify and observe different liquid samples [9,10,11]. The sensing occurs when the solution undergoes interaction with propagating or resonating modes of electromagnetic waves. Due to this interaction, the complex permittivity of the material changes, and it manifests itself as a frequency change, attenuation, reflection of the signal, or a phase shift. The sensing principle relies on the changes of the device response in terms of the transmitted power, resonant frequency, and 3-dB bandwidth. Among these techniques, spectroscopy is a commonly used technique to measure the permittivity of liquid samples as a function of frequency that gives information about the relative permittivity and the loss tangent of the sample [12]. It is being used to distinguish between materials, but the sensitivity is not high and can sense the variation of permittivity values from 3 to 11 for resonant frequency change of 5.2 GHz to 4.1 GHz [13]. As the gasoline sensor needs to be small, microwave technologies help to make the sensor compact and more configurable than mass spectrometry or gas chromatography.

Among microwave-based designs, resonant cavities are easy to fabricate, require a minimal sample, and possess excellent resonant characteristics that ensure higher accuracy [14,15] and high sensitivity [16] when used explicitly in sensing resonant frequency locations. This attribute is useful for different material characterization, including nutrient monitoring in waste water treatment [17], food evaluation, and analysis [18]. Note that the resonant frequency is comparatively less sensitive to the impurities that increase the conductivity of fuel blends [19]. Many researches are based on cavity designs. Guo et al. in [14] used cylindrical cavity opened at both ends for online water content measurement by measuring resonant frequency of the cylindrical cavity. Another reported sensor utilizes a cylindrical cavity for resonant frequency change between 5 and 5.7 GHz to analyze a two phase gas–liquid flow regime in a pipeline [20]. Ethanol content present in gasoline has a high permittivity that allows useful sensors to be built using relatively short lengths of metal enclosure cavities [19].

In this paper, a microwave-based sensor is proposed with high sensitivity and accuracy for the distinction of various gasolines. The difference of permittivity values between different gasoline is extremely minor ranging from 0.05 to 0.15, which requires high sensitivity. The sensor is optimized to have a specific resonant frequency according to the unique permittivity of the each gasoline sample; the unknown sample can be found by comparing this resonant frequency with known samples. For the sensor design, a one port cylindrical cavity sensor is adopted, which results in small size and low cost. ANSYS high-frequency structure simulator (HFSS) and a vector network analyzer (VNA) are used to verify the design and performance of the proposed cylindrical cavity sensor. The novel contribution of the presented sensor is its robustness with temperature variations while maintaining sensitivity and accuracy.

The content of the paper is organized as follows. Section 2 presents the proposed sensor modeling. Section 3 presents the design procedure and simulation analysis performed by HFSS. In Section 4, the sensor fabrication and experimental setup is described followed by the performance evaluation of the proposed sensor, on the basis of simulation and experimental results, is discussed. The conclusion of the paper is presented in Section 5.

## 2. Cylindrical Cavity Sensor Modeling

The proposed sensor model is shown in Figure 1. This sensor can be separated into two components: a monopole and a cylindrical cavity which can be built from circular waveguide by shorting metal walls at both ends. Figure 1a illustrates the geometrical structure of the cylindrical cavity resonator, which consists of closed copper cylinder filled with sample of permittivity ϵ, where $\epsilon =\epsilon_{0}\epsilon_{r}$ ($\epsilon_{0}$ is the permittivity of free space approximately equal to 8.85 × 10^−12^ F/m, and $\epsilon_{r}$ is the relative permittivity of the fuel). The proposed sensor design is chosen to be a closed metal cavity structure to avoid any external interaction since the sensor is to be used either inside the fuel tank or at the fuel inlet of the car where there is a chance of the presence of other electromagnetic waves. The interior of the cavity is a hollow uniform cylindrical shape. A small hole is kept at the top center of the cylindrical cavity for providing the feed source. The feed source is vertically placed into the cavity through the hole which acts as an intermediator from external equipment to the cavity for transmitting microwaves inside the metal enclosure and for monitoring its resonant behavior. A monopole has been chosen as a feed source as shown in Figure 1b. The degree of coupling is controlled by adjusting the length of the probe which is kept small to have minimum interaction with the field inside. The monopole is coated with thin layer of insulating medium Teflon to avoid direct contact with the testing gasoline samples, so as to avoid corrosion, which may lead the sensor to operate abnormally. Even though the current in the monopole is very small, an electric field is created between monopole and adjacent wall of the resonator, which is maximum at the location of monopole and also perpendicular to the wall. The current also generates a magnetic field that radiates like a magnetic dipole tangential to the wall.

The resonant frequency of the cavity exhibits a substantial change with a small change in square root of permittivity ($\sqrt{\epsilon }$) of gasoline samples. A basic experimental schematic utilizing the proposed sensor for measuring permittivity of different gasoline samples is shown in Figure 2.

When resonance occurs, there is the possibility of the occurrence of various electromagnetic modes within a specific cavity, where different modes have their own resonant frequencies, dimensions of cavities, and associated Q-values [21]. The basic modes for cylindrical cavity resonators are transverse magnetic, *TM*_01_, and transverse electric, *TE*_01_ and *TE*_11_ [22]. In microwave theory, the higher-order mode decays rapidly, therefore *TM*_01_, the first higher mode of the circular waveguide, was selected. This selection was made since it is easier to manufacture as compared to the TE mode in actual application environment. In cylindrical cavity, the wavelength λ of the *TM*_01_ mode can be defined as [23]
(1)λ=2.612a
where *a* is the internal radius of the cavity. The resonant frequency for a given mode is calculated using Equation (Equation 2), where $f_{nmt}$ is the resonant frequency of the *TM_nmt_* mode; *m*, *n*, and *t* are the number of full-wave patterns along the circumference, the number of half-wave patterns along the diameter, and the number of half-wave along the height of the cylindrical cavity, respectively; *d* is the height of the cavity; and $P_{nm}$ is the *m*th root of Bessel function of *n*th order. The values of $P_{nm}$ are listed in Table 1; c=299,792,458 m/s is the speed of light, $\mu_{r}$ is the relative permeability (value considered to be 1 since the gasoline samples are non-magnetic in nature), and $\epsilon_{r}$ is relative permittivity of the filled material inside the cylindrical cavity sensor.
(2)$$f_{nmt}=\frac{c}{2\pi \sqrt{\mu_{r}\epsilon_{r}}}\sqrt{{\left(\frac{P_{nm}}{a}\right)}^{2}+{\left(\frac{t\pi }{d}\right)}^{2}}$$
From Equation (Equation 2), it can be noticed that the utilization of higher frequencies may lead to small size cavities that in turn need a much lower quantity of the sample.

## 3. Simulations Results of the Proposed Sensor

For transmitting and receiving electromagnetic waves, more than one antennas can be positioned inside the cavity. The most common arrangements are one-port or two-port cavities [24]. For one-port arrangement, when the testing material inside the cavity interacts with electromagnetic waves, a part of the power is reflected back which is termed the reflection coefficient or the return loss that can be measured with the help of VNA. This measure is used in the following simulations and experiments to analyze the performance of the proposed sensor. For sensitivity analysis, resonant frequency separation is chosen for the distinction of gasoline samples.

### 3.1. Design of the Proposed Sensor

The proposed sensor is designed to attain wide frequency separations and low reflection coefficients for all scenarios. The design steps of proposed cavity sensor are as follows.
Selection of permittivity (ϵ): First, we selected the permittivity of gasoline for simulating the sensor model. The gasoline samples used in our experiments are high driveablity index (HiDI) samples with average permittivity value of 2.1 and the permittivity of normal gasoline used in the design of this sensor is 2.15, which is similar to the average permittivity of HiDI gasoline.Selection of the cylindrical cavity mode and design frequency: The procedure of selecting TM_012_ and the design frequency is already described in Section 2.Selection of cylindrical height (*d*): We have chosen the height of the cylindrical cavity through simulations by sweeping it for the frequency range of 6.7 to 7.7 GHz. Figure 3a shows the optimization process of selecting the cylinder height. It is chosen to be 35 mm as it gives the minimum reflection coefficient for the given frequency range.Selection of monopole height (*h*): To find the best height for the design, we have observed that the height of 8 mm of monopole gives good response in terms of lower reflection coefficient, as shown in Figure 3b.


Similarly, all those optimized parameters were chosen as they gave the best results in frequency separations and reflection coefficients.

The selected design parameters for the cavity resonator with height of 35 mm, radius of 19.2 mm, and feeding height of 8 mm showed minimum reflection coefficient and largest frequency separation. The optimized parameters for the designed sensor are given in Table 2.

### 3.2. Simulations of the Proposed Sensor

The electromagnetic field of cylindrical cavity sensor with *TM*_012_ mode is simulated using HFSS software in Figure 4a,b. It can be observed from these figures that the magnetic field of the TM mode is parallel to the cavity bottom and tangent to the wall of the cylinder, and the electric field is perpendicular to both the magnetic field and the wall at the location of the probe. The intensity of electric and magnetic field is maximum at the center of the cavity and decreases approaching from the cavity center to the boundary. Therefore, these simulation results are in good agreement with the theory in Section 2.

By using HFSS, simulations have been conducted to analyze how the chosen design parameters and selected *TM*_012_ mode for the presented sensor affect reflection coefficients and frequency separations for different gasolines. We considered five different gasoline samples among samples specified in Table 3, which represents different types of standard gasoline/ethanol blends according to their ethanol content and emissions of nitrogen oxides (NOx), total hydrocarbon (THC), non-methane hydrocarbon (NMHC), and carbon monoxide (CO). The reflection coefficients of the selected five samples of LEV3, Tier3, Euro4, Cold CO, Tier2 and one assumed sample (permittivity differ by 0.01 with LEV3) are shown in Figure 5 for a frequency range from 6.6 GHz to 7.7 GHz. Note that Euro4 of 2.134 permittivity has a high quality factor, as it has closer permittivity to the designed permittivity 2.15. On the other hand, Tier3 of 2.289 permittivity has a low quality factor since its permittivity is not close to the designed value and has very small complex permittivity compared to the other samples. The resonant frequency separation of four samples is greater than 100 MHz and the reflection coefficients are below −12 dB. The minimum frequency separation of 18 MHz lies between the assumed sample and LEV3, which is still possible to distinguish.

Next, we simulated three different fuels, such as kerosene, gasoline, and heavy oil as mentioned in Table 4, to verify the resonance performance of the proposed sensor. The reflection coefficient according to frequency is shown in Figure 6. The response is plotted for a frequency range of 6 GHz to 8.5 GHz. Note that the proposed sensor exhibits remarkable resonant frequency separation between different fuel samples and has low reflection coefficients below −31 dB. Also, it can be easily observed that the resonant frequency has a negative relationship with permittivity.

## 4. Experimental Results and Discussion

### 4.1. Sensor Fabrication and Experimental Setup

The proposed cylindrical cavity sensor is fabricated using conductive copper cylinder of inner radius (*a*) 19.2 mm and height (*d*) 35 mm. The height *h* of the probe is set to 8 mm, which is equal to one-quarter of the wavelength of frequency 7.119 GHz, making the input impedance nearly equivalent to that of an open circuit. The dimensions of coupling probe are 0.5 mm × 8 mm, and it is wrapped with Teflon with radius size of 3 mm as shown in Figure 7b. The manufactured sensor is shown in Figure 7a. In this model, a coupling probe of height *h* and radius *r* is designed by extending the feeding coaxial cable, with a small distance, into the cylindrical cavity sensor and fixing at the bottom center.

The experimental set-up consists of a cylindrical cavity resonator with monopole, VNA, external pump, thermostatic bathing, cable, capillary tubes, and gasoline samples. The cylindrical cavity sensor is connected through a connector to supply microwave signals and to allow measurements to be taken using a calibrated series network analyzer VNA (100 KHz–8.5 GHz E5063A, KEYSIGHT) via a 50 Ω coaxial cable. The gasoline solutions are introduced in the cavity via a thermoplastic capillary. In particular, the capillary is connected to an external pump via tube fittings, so that a continuous gasoline flow is created and no air bubbles are formed in the capillary. A thermostatic bath and thermometer are used to control and monitor the temperature of gasoline samples as shown in Figure 8b. The experimental setup described above is shown in Figure 8a with gasoline bottles with different permittivities and microwave cavity sensor.

### 4.2. Results and Discussion

Two important properties of dielectric materials are complex permittivity $\epsilon^{*}=\epsilon -j\epsilon^{\prime \prime }$ and loss tangent $tan\delta =\frac{\epsilon^{\prime \prime }}{\epsilon }$. The real part, ϵ, of $\epsilon^{*}$ is the permittivity (i.e., product of the free space permittivity $\epsilon_{0}$ and the relative real/absolute permittivity $\epsilon_{r}$) quantifying the stored energy within the medium and the imaginary part $\epsilon^{\prime \prime }$ represents dielectric loss factor related to the dissipation of energy within the medium. The ratio tanδ quantifies the loss of power due to the propagation in a conductor. To analyze this loss of power, we performed experiments with the proposed sensor on gasoline samples with different permittivities as shown in Figure 9 using Table 5. The complex permittivity values were measured using N1501A permittivity measuring kit. The temperature of 20 °C was chosen as the proposed sensor is designed at the same temperature and also the N1501A kit can measure permittivity only from 0 to 120 degrees.

The permittivity of a solution changes with change in temperature, which in turn changes the resonant frequency. However, the effects are relatively small for hydrocarbon lubrication oils. The typical decrease in permittivity for hydrocarbon oils is about 0.0013 to 0.05 percent per degree Celsius [27]. To verify this relationship and to analyze the ability of the proposed sensor for various gasoline samples in cold condition, we evaluated the sensor response as shown in Figure 10. The temperature range of 0 °C to −35 °C was chosen for the cold condition, and the measurements were made with the help of thermostatic bath. In Figure 10, small and linear change in resonant frequencies according to the temperature variation can be observed. The minimum frequency separation of 31 MHz lies between Euro4 and Cold CO at 0 °C temperature. The resonant frequency shifts towards a slightly higher value with the rise in temperature for Euro4, Cold CO, and Tier2. For Tier3 and LEV3, the resonant frequency shifts slightly to a lower value, as these two gasoline samples have higher percentages of ethanol and the ethanol tends to have low permittivity at higher temperatures [28]. Nonetheless, the performance of the proposed sensor is almost the same which confirms the robustness of the sensor with temperature changes.

Using optimized parameters, we performed full wave analysis of the proposed sensor, using HFSS for the frequency range of 6.5 GHz to 7.5 GHz under different temperature conditions. By sweeping frequency from 6.5 GHz to 7.5 GHz, we achieved reflection coefficient for LEV3, Tier3, Euro4, Cold CO, and Tier2 at −35 °C to 0 °C temperatures, as shown in Figure 11a,b, respectively. The proposed sensor fulfills the requirement of large frequency separations between different samples making it sensitive towards small changes in permittivity of a given sample. The gasoline sample, Tier2 shows minimum reflection coefficient at −35 °C with −63.64 dB dip at a resonant frequency of 7.15 GHz. In Table 6, the frequency separations of close resonant peaks are listed. At higher temperatures, the resonant frequency increases. Higher temperatures increase the activity of electrons and reduce the relative permittivity of fuels; thus, resonant frequency as well as reflection coefficient increases. Here, note that the fuel with lower permittivity has higher resonant frequency and vice versa, i.e., Tier 2 as the fuel with the lower permittivity and LEV 3 as the fuel with higher permittivity. These results reveal that the proposed cylindrical cavity sensor is still very sensitive to distinguish normal and HiDI gasoline under different temperature conditions.

The relationship between permittivity of the different gasoline samples and resonant frequency is shown in Figure 12a, where curve 1 shows theoretical values and curve 2 shows experimental values. The resonant frequencies of theoretical data are slightly higher than experimental values. The qualitative analysis of Figure 12a suggests that resonant frequency decreases monotonically with increase in permittivity in both curves. The experimental results are in agreement with that of theoretical results. Now, to perform quantitative analysis of the given data, we define percentage error as follows; Percentage Error = |Experimental values − Theoretical|/Experimental values × 100. Figure 12b represents the percentage error between the theoretical and experimental values of resonance frequencies for various gasoline samples with different permittivity. The overall error is below 1.7%, which confirms the accuracy of the proposed sensor.

All the measurements were repeated five times with the same sample, and their average value was considered. Samples with different dielectric constants in experimental measurements show various quality factors. However, as the repeated measurement error of all samples is 0.5 MHz, which is smaller than the minimum resonant frequency difference 29 MHz between samples, it is possible to distinguish between different samples even considering LEV3 with a lower quality factor.

Finally, we have performed a comparison of our proposed method with the capacitor method, which is one of the DI measurement method presented in [5]. The results are summarized in Table 7. The sensitivities of the proposed sensor are achieved by taking the ratio of difference in resonant frequencies (fr) of the samples to the difference in DI values and listed in Table 7a. Similarly, sensitivities of the sensor in [5] were calculated by taking the ratio of voltage difference of the samples to the difference in DI values, and they are listed in Table 7b. It can be seen that the proposed sensor is better in terms of sensitivity to DI.

## 5. Conclusions

A cylindrical cavity sensor is proposed to distinguish between different DI gasoline for different temperature conditions. The sensing principle relies on the change of resonant frequency caused by the change of permittivity of fuel samples inside the cavity sensor. The proposed sensor utilizes a circular cavity design, which is low-cost, compact, easy to fabricate, easy to calibrate, and sensitive in terms of resonant frequency separation. The performance of the proposed sensor design was verified by HFSS simulations and experiments. The resonant mode chosen for the waveguide is *TM*_012_ for easier manufacturing as compared to TE. The presented sensor is fabricated with an enclosed conductive copper cylinder and a fixed coupling probe wrapped with Teflon mounted inside the top center of the cavity resonator. The experimental results are in agreement with theoretical results, which showed the accuracy of the proposed sensor. The novel contribution of the presented sensor is its robust behavior under temperature variations and still achieving reasonable resonant frequency separation for different gasoline samples with tiny difference in permittivities. Therefore, we conclude that the presented sensor can be effectively used to distinguish gasoline with different DI and has potential applications of liquid classification.

## Figures and Tables

**Figure 1 sensors-19-04626-f001:**
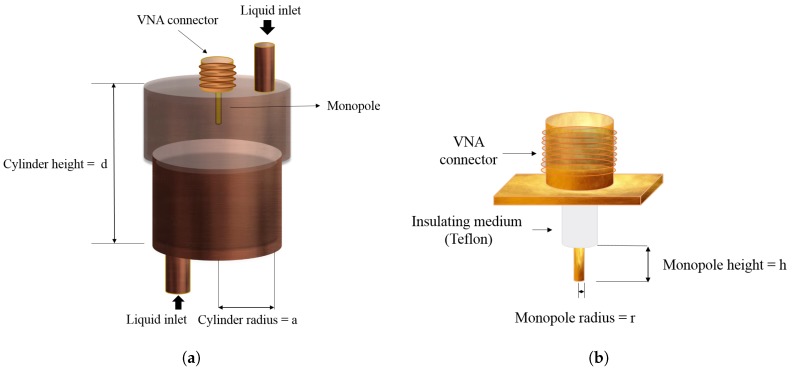
Geometrical structure of (**a**) cylindrical cavity resonator. (**b**) A coupling probe (monopole) fixed at the top center of cavity.

**Figure 2 sensors-19-04626-f002:**
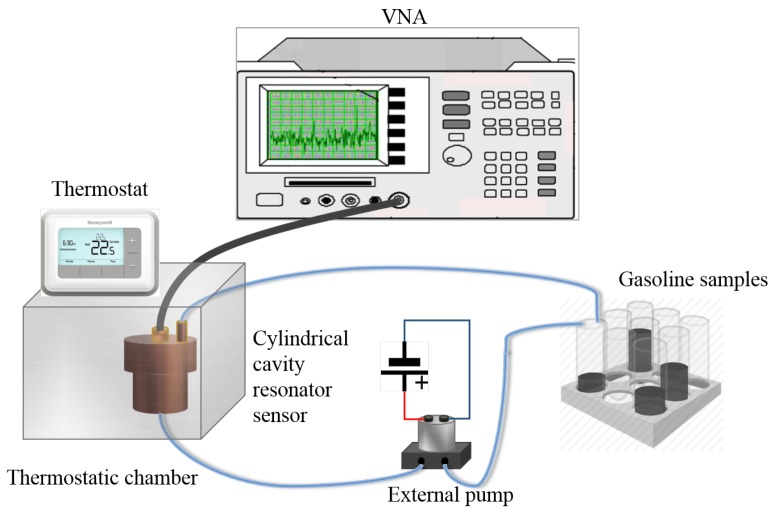
Experimental schematics of microwave cavity resonator for measuring permittivity of different gasoline samples.

**Figure 3 sensors-19-04626-f003:**
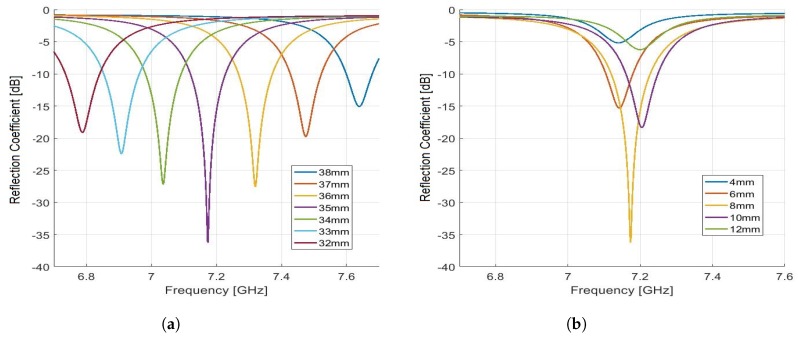
(**a**) Simulation results for the optimization of cylindrical cavity height (*d*). (**b**) Simulation results for the optimization of monopole height (*h*).

**Figure 4 sensors-19-04626-f004:**
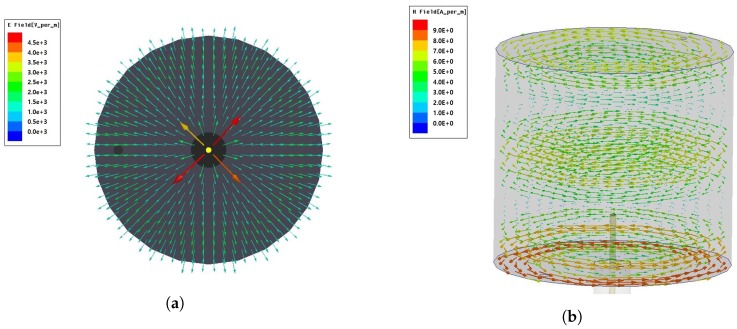
(**a**) Top view of electric field. (**b**) Side view of magnetic field.

**Figure 5 sensors-19-04626-f005:**
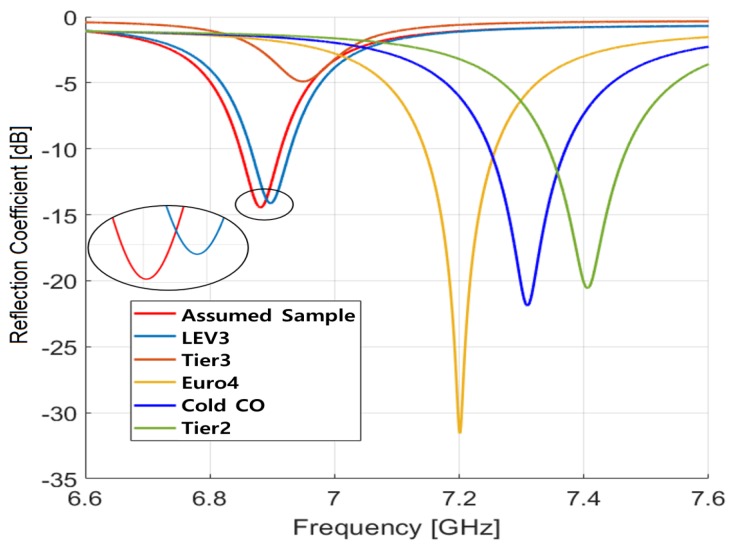
Simulation results of resonant frequency for the proposed sensor at 20 °C temperature for various gasoline samples.

**Figure 6 sensors-19-04626-f006:**
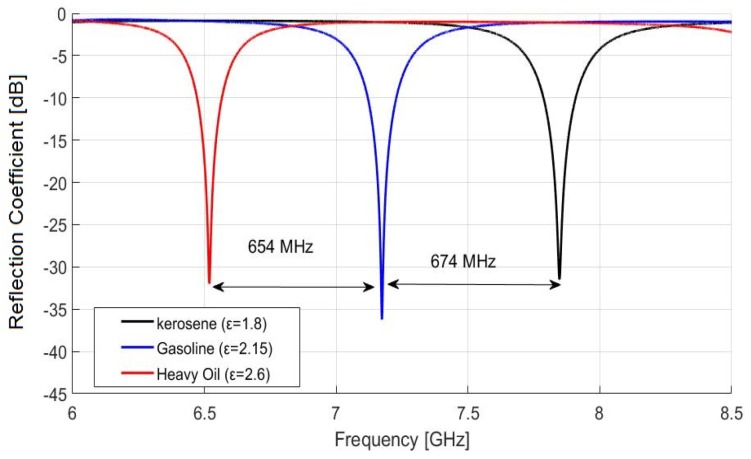
Reflection coefficient versus resonant frequency of the proposed sensor for kerosene, gasoline and heavy oils at room temperature (20 °C).

**Figure 7 sensors-19-04626-f007:**
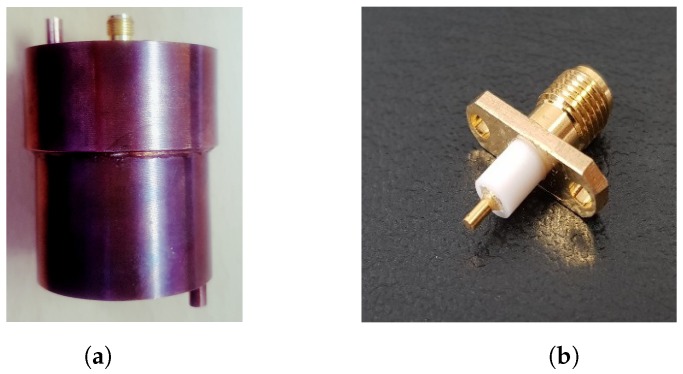
(**a**) Actual fabricated sensor. (**b**) Coupling probe (monopole).

**Figure 8 sensors-19-04626-f008:**
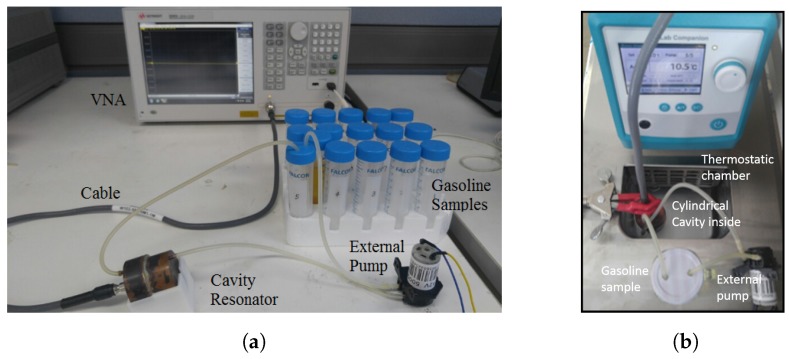
(**a**) Original experimental set-up. (**b**) Thermostatic bathing.

**Figure 9 sensors-19-04626-f009:**
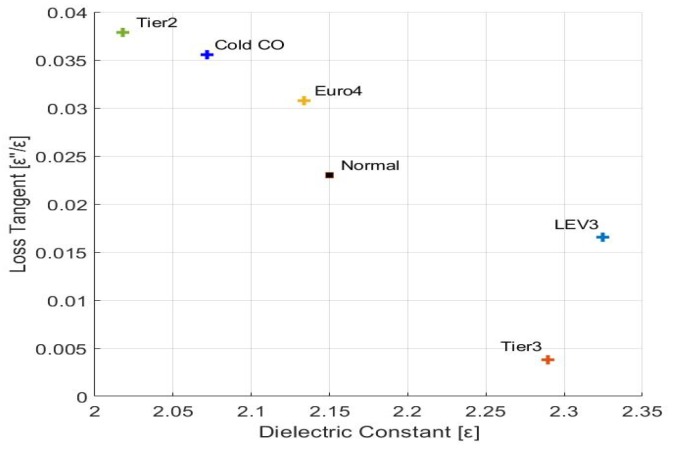
Variation of loss-tangent (tanδ) with permittivity (ϵ) for different gasoline samples.

**Figure 10 sensors-19-04626-f010:**
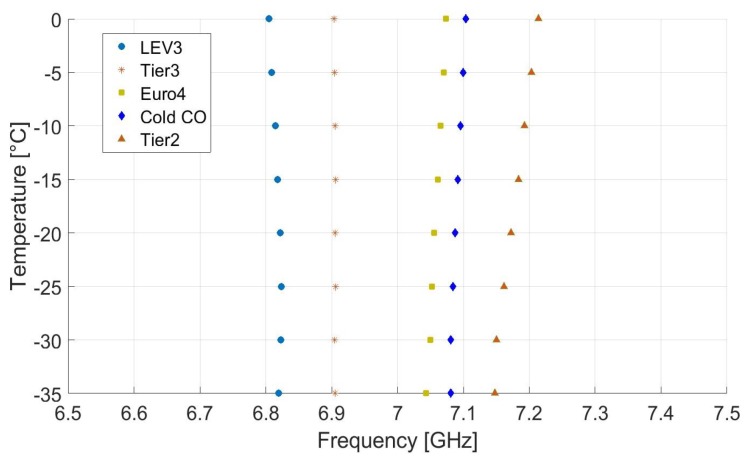
Experimental results of resonant frequency for the proposed sensor at different temperature for various gasoline samples.

**Figure 11 sensors-19-04626-f011:**
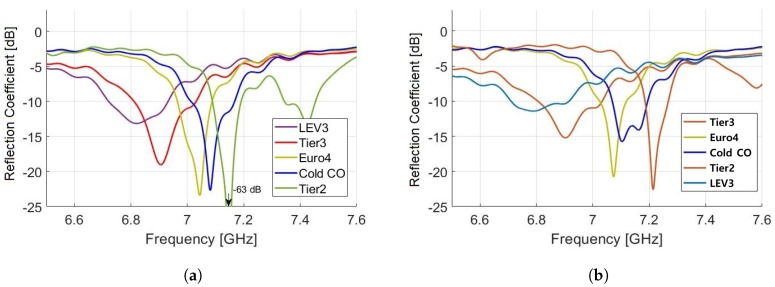
Reflection coefficient analysis of the proposed sensor at (**a**) −35 °C and (**b**) 0 °C.

**Figure 12 sensors-19-04626-f012:**
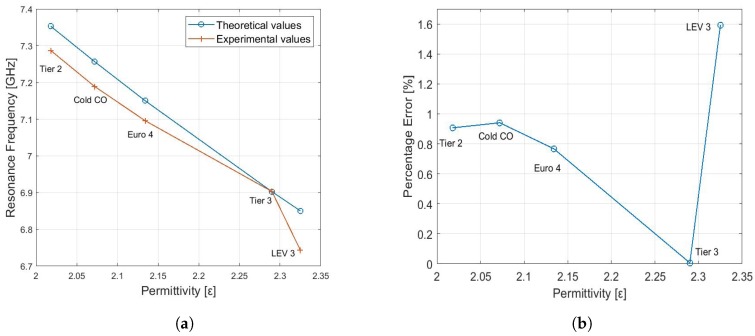
(**a**) Theoretical and experimental results of the proposed sensor for different gasoline samples at 20 °C temperature. (**b**) Percentage error between the theoretical and experimental results of the proposed sensor for different gasoline samples at 20 °C temperature.

**Table 1 sensors-19-04626-t001:** The roots ($P_{nm}$) of the Bessel function.

	m	1	2	3
n				
0		2.405	5.520	8.654
1		3.832	7.016	10.174
2		5.135	8.417	11.620

**Table 2 sensors-19-04626-t002:** Design parameters of the proposed sensor.

Parameter	Dimension (mm)	Parameter	Value
*a*	19.2	$P_{01}$	2.405
*d*	35	t	2
*h*	8	Permittivity (ϵ)	2.15
*r*	0.5	mode	TM

**Table 3 sensors-19-04626-t003:** Gasoline samples according to emission standards for light vehicles, g/km [25].

Index	Gasoline Samples	CO	THC	NMHC	NOx	Ethanol (vol.%)
1	Tier2 (Indolene)	3.5	-	2.16	4.44	<10
2	Tier3 (Indolene E10)	3.5	-	-	4.9	14.8–15.2
3	LEV2 (Phase 2)	6.4	0.032	-	0.05	<10
4	LEV3 (Phase2 E10)	-	-	-	0.07	9.75–10.25
5	Cold CO	1.7	0.45	-	0.17	<10
6	Cold CO E10	1.5	0.4	-	0.2	10
7	Euro4	1	0.10	-	0.08	<10
8	Euro5	1	0.10	0.068	0.06	<10

**Table 4 sensors-19-04626-t004:** Simulation results of proposed sensor showing resonant frequency, reflection coefficient and permittivity for kerosene, gasoline, and heavy oils [26] at room temperature (20 °C).

	Kerosene	Gasoline	Heavy Oil
$f_{r}$ (GHz)	7.847	7.173	6.519
Γ (dB)	−31.5	−36.2	−32.0
ϵ	1.8	2.15	2.6

**Table 5 sensors-19-04626-t005:** Real permittivity and measured complex permittivity for different gasoline samples at 20 °C.

Gasoline Sample	Permittivity Real (ϵ)	Permittivity Complex ($\epsilon^{\prime \prime }$)
Tier2	2.018	0.0764
Cold CO	2.072	0.073
Euro4	2.134	0.066
Normal gasoline	2.15	0.0494
Tier3	2.289	0.0087
LEV3	2.324	0.0385

**Table 6 sensors-19-04626-t006:** Frequency separation of various gasoline samples at different temperatures.

Gasoline Samples	Resonant Frequency [GHz]	Reflection Coefficient [dB]	Frequency Separation [MHz]
LEV3 (−35 °C)	6.818	−13.14	86
Tier3 (−35 °C)	6.904	−19.02	
Euro4 (−35 °C)	7.042	−23.4	36
Cold CO (−35 °C)	7.078	−22.69	
LEV3 (0 °C)	6.791	−11.41	112
Tier3 (0 °C)	6.903	−15.21	
Euro4 (0 °C)	7.074	−13.36	29
Cold CO (0 °C)	7.103	−15.71	
Tier2 (−35 °C)	7.148	−63.64	67
Tier2 (0 °C)	7.215	−22.37	

**Table 7 sensors-19-04626-t007:** Sensitivity/DI: (**a**) proposed RF method and (**b**) capacitor method.

**Sample**	**DI**	**Resonant Frequency** $f_{r}$ **[MHz]**	**Sensitivity [100 KHz/DI]**
Cold CO	1132	7191.0	51.05
Tier3	1075	6900.0	
LEV3	1168	6788.9	11.93
		(**a**)	
**Sample**	**DI**	**Resonant Frequency** $f_{r}$ **[mV]**	**Sensitivity [100 mV/DI]**
A	1131	2180	0.020
B	1087	2090	
C	1164	2230	0.018
		(**b**)

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
