# Peer review of "Cylindrical Cavity Sensor for Distinction of Various Driveability Index Gasoline with Temperature Robustness"

_sensors, 2019, doi:10.3390/s19214626_

Round 1

Reviewer 1 Report

Author Should justify the application of Cylindrical Cavity Sensor to calculate Gasoline Driveability Index for the Multi fuel (Diesel, Biogas and hydrogen) operated CI engine? How the Radio frequency (RF) and microwave techniques are cost effective rather than to analyzed liquid samples using high resolution gas chromatography/mass spectrometry (HRGC/MS)? Author should give a parallel comparison. Author should justify that Cylindrical Cavity Sensor is applicable inside the Cylindrical where we found very high temperature. There are some typographical and grammatical errors in manuscript, please go through them during the re-submission. References [7] [8] [9] are not papers related to gasoline. Please explain why you think that the radio frequency (RF) and microwave technology is strong and cost-effective for this gasoline distinction experiment? In Section 3.1, how did you get the conclusion "those parameters were chosen that gave best results in frequency separations and reflection coefficients", in the absence of a control experiment? The labels in Figure3.(a) and (b) are too small to be seen after being printed. Please adjust. There are some grammatical errors, please correct.

Author Response

Dear Reviewer

I would like to thank for your valuable comments.

By following your comments, I attached response file so that you can check our replies according to your comments.

I appreciate your help and suggestion in advance.

Best regards,

Prof. Chong Lee

Reviewer 2 Report

This paper addresses a typical problem of detecting liquid samples (different types of fuel in this case) to identify them using a method of microwave resonant cavities in a given frequency range (6.6Ghz to 7.7HGz).

To do this, in this work, different types of fuel are classified based on the change in resonance frequency produced by the changes in the resonant modes permittivities in a cylindrical cavity. One of the main achievements of this work is to identify these changes, when the ranges of permittivity that are handled are so small (0.4 to 0.15).

In addition, the theoretical study and the design carried out in simulation platforms (HFSS) have been validated in an experimental installation with very robust results, taking into account the small dimensions of the developed sensor and that the fuel must be continuously circulated in the cavity.

From the point of view of the scientific method, a study consistent with the sensor design is presented, which allows validating the results of several fuel samples with a high degree of success. However some clarifications should be made to corroborate the study.

First, it would be important that authors give comparative numerical details of the sensitivity measurement of this sensor with respect to traditional methods. On the other hand, it would have been desirable to test with other fuel samples (Figure 4), so that the sensitivity in the classification can be verified, when the BWs are below 30MHz.

In addition, Figures 4 and 5 should be grouped. Are the same resonant modes what are measured in each case?

Another important aspect is to clarify why the study of the samples is done at temperatures of 0ºC and -35°C and next, the experimental measurement are carried out at 20°C. How is the validity of that result technically justified?

Author Response

(The authors gave the same response as above.)

Reviewer 3 Report

The topic of this article is very interesting and important, but the text itself causes a number of questions.

- It is not clear how the authors calculated the loss tangent, which is shown in Fig. 8. From the text we can conclude that only the reflection coefficient was measured. It is not clear how the loss tangent could be calculated from the reflection coefficient measurements. From only the reflection coefficient, such information is difficult to extract.

- The measurement accuracy of the resonant frequency at different permittivity values is also questionable. As I understand, the change of permittivity will change not only the resonance frequency, but also the loaded quality factor of the resonator (which, in turn, also will shift the resonance frequency). This situation is confirmed by blurring of the peaks of the reflection coefficient. Authors should try to separate these two factors one from another.

- The sentence in lines 163-164 is not clear: "Also, it can be easily observed that resonance frequency has a negative relationship with permittivity and the resonance frequency". Apparently, the resonance frequency is negatively interconnected with itself.

- Figures 2 and 7a have the same sense. Do the authors really need to demonstrate both figures?

Author Response

(The authors gave the same response as above.)

Reviewer 4 Report

Line Comment
1 There is a "1".
6 Why are you using just a real number for the "normal gasoline" permittivity?
8 The cavity is not limited to a single resonant frequency. You should include the mode if you are going to specify a frequency.
8 The radius is 19.235mm. Is this level of precision required for acceptable performance?
22 Should the word "sum" really be "function"?
28 It would read better if you included the authors' names for reference [5].
31 You do not need the word "sensors".
31 The use of both "out" and "exhaust" seems unnecessary. Exhaust would be preferable in both instances.
36 Reconsider the word "strong"
37 "In peculiar" is not correct. You may have intended to write "In particular", but that does not follow from the previous sentence, and it would be better to delete it.
39 The permittivity does not change, the speed of light in the sample volume changes.
41 Change "resonance frequency" to "resonant frequency" throughout the paper.
43 Change "perform distinction of" to "distinguish between".
52 See comment for line 28.
57 The claim that this is a novel sensor is not credible. Any novelty here lies in the application.
59 Is the 0.04 and 0.15 for the "minor" difference 4% and 15%? Where do these figures come from?
61 "out" should be deleted.
68 This sentence should be rewritten. "??" should be "4".
69 Remove "At the end".
74 "shortening" should be "shorting".
76 Use the multiplication symbol (×) not the letter x. There are several other instances.
77 Use "external interaction" rather than "interaction from outside"
85 Remove the sentence, "The length …"
86-89 Merge into one sentence.
92 "of the loop"?
92 The last sentence in this paragraph is not needed.
96 It would be useful to include the safety measures used for handing gasoline in these experiments.
102 The brackets should also enclose TE11.
107 "operation frequency of the cylindrical cavity resonator" should be changed to "resonant frequency of the TMnmt mode".
110 remove "for the TE or TM modes of the cavity resonator". The roots of the Bessel function (and its derivative) are independent of their application.
116 Refer to the equation as Equation 2.
116-117 The claim that the cavity volume is inversely proportional to the square of the resonant frequency is wrong. The same volume can give different frequencies.
116-117 The claim that the pnm and t are directly proportional to the square of the resonant frequency is wrong. Also changing Pnm to a different root would change the mode.
137 The 7.124GHz is the calculated value for TM012 if c = 300 000 000m/s. The exact value of c is 299 792 458m/s which gives afrequency of 7.119GHz.
140 Providing the circumference serves no purpose, and if it were required it would be rounded up to 120.86mm
142 Not "TM" but "TM012".
152 nitrogen oxides (NOx)
154 "sweeping" not "swiping".
154 What complex permittivity values were used for this simulation?
155 No full stop.
159 Where does the permittivity data come from for Table 4. There are no axes labels for Figure 5.
175 Finish the sentence after "cavity sensor".
181 Change "is" to "represents".
183 Power propagating in a conductor?
190 The sentence beginning, "This analysis.." should be removed.
199 remove ×109 from Figures 9 and 10. The units are GHz.
200 How was HFSS used to simulate different temperatures?
203 Remove the sentence, "It can be …"
203 What are the extra modes in Figure 10?
207 "dip point"?
220 Equation 3 is too basic to include.

The missing information regarding loss tangent values and the temperature effects on permittivity, that would be required to perform a simulation, are major concerns.

You need to explain why the capacitor method (reference 5) is not a better approach than the one proposed. 

There are a lot of missing "a"s and "the"s.`

Author Response

(The authors gave the same response as above.)

Round 2

Reviewer 3 Report

Authors have revised the paper successfully: it can be now considered for publication in the present form.

Author Response

Dear Reviewer

We would like to thank for your close reading and valuable time spent on our manuscript so far. We have revised the manuscript complying with other reviewers' comments and corrected grammatical mistakes. 

We appreciate your help for improving quality of our paper.

Best regards,

Prof. Chong Lee

Reviewer 4 Report

The paper is much improved but there are still a number of issues.

Author Response

Dear Reviewer

We would like to thank for your close reading and valuable time spent on our manuscript. We have revised the manuscript complying with reviewers' comments.

In the revised manuscript, modified or newly added sentences are printed in bold face and underline so that you may check them.

We appreciate your help for improving quality of our paper.

Best regards,

Prof. Chong Lee